# The Role of Ultrasound in the Evaluation of Inguinal Lymph Nodes in Patients with Vulvar Cancer: A Systematic Review and Meta-Analysis

**DOI:** 10.3390/cancers14133082

**Published:** 2022-06-23

**Authors:** Debora Verri, Francesca Moro, Simona Maria Fragomeni, Drieda Zaçe, Sonia Bove, Federica Pozzati, Benedetta Gui, Giovanni Scambia, Antonia Carla Testa, Giorgia Garganese

**Affiliations:** 1Gynecology and Breast Care Center, Mater Olbia Hospital, 07026 Olbia, Italy; debora.verri@materolbia.com (D.V.); sonia.bove@materolbia.com (S.B.); giorgia.garganese@materolbia.com (G.G.); 2Unità di Ginecologia Oncologica, Dipartimento Scienze della Salute della Donna, del Bambino e di Sanità Pubblica, Fondazione Policlinico Universitario A. Gemelli IRCCS, 00168 Rome, Italy; francesca.moro@policlinicogemelli.it (F.M.); federica.pozzati@guest.policlinicogemelli.it (F.P.); giovanni.scambia@policlinicogemelli.it (G.S.); antoniacarla.testa@policlinicogemelli.it (A.C.T.); 3Dipartimento Universitario Scienze della Vita e Sanità Pubblica, Università Cattolica del Sacro Cuore, 00168 Rome, Italy; drieda.zace@unicatt.it; 4Dipartimento Diagnostica per Immagini, Radioterapia Oncologica ed Ematologia, Fondazione Policlinico Universitario A. Gemelli IRCCS, 00168 Rome, Italy; benedetta.gui@policlinicogemelli.it

**Keywords:** lymph nodes, groin, ultrasound, vulvar cancer

## Abstract

**Simple Summary:**

Currently, around 30% of vulvar cancer cases at first diagnosis are spread to the inguinal lymph nodes. Preoperative staging of patients affected by vulvar carcinoma is still a hot topic. To date, MRI has shown a great diagnostic accuracy on defining disease extension to soft tissue and deep organs. At present, regarding the study of inguinal nodes, the PET/CT scan has shown a high negative predictive value, although in the presence of a suspicious/positive report it should be taken with caution. We report the results of a study aimed to investigate the role of groin ultrasound in the assessment of lymph nodal status in vulvar cancer. Furthermore, this review represents the most accurate collection of papers available in the literature. This work demonstrates that groin ultrasound can be considered a valuable tool for risk assessment of the presence of groin lymph node metastases. Achieving a high diagnostic accuracy would allow tailored surgical planning with access to minimally invasive surgery techniques for an increasing number of patients.

**Abstract:**

Objective. To determine the efficacy of ultrasound in assessing the inguinal lymph nodes in patients with vulvar cancer. Methods. A systematic review of published research up to October 2020 that compares the results of ultrasound to determine groin node status with histology was conducted. All study types that reported primary data on the role of ultrasound in the evaluation of groin lymph nodes in vulvar cancer were included in the systematic review. Data retrieved from the included studies were pooled in random-effects meta-analyses. Results. After the screening and selection process, eight articles were deemed pertinent for inclusion in the systematic review and meta-analysis. The random-effects model showed a pooled Se of 0.85 (95% CI: 0.81–0.89), Sp of 0.86 (95% CI: 0.81–0.91), PPV of 0.65 (95% CI: 0.54–0.79) and NPV of 0.92 (95% CI: 0.91–0.94). There was a pooled LR+ and LR− of 6.44 (95% CI: 3.72–11.4) and 0.20 (95% CI: 0.14–0.27), respectively. The pooled accuracy was 0.85 (95% CI: 0.80–0.91). Conclusions. Although the studies had small sample sizes, this review represents the best summary of the data so far. Ultrasound has revealed high sensitivity and high negative predictive value in the assessment of nodal status in vulvar cancer.

## 1. Introduction

Vulvar cancer is a rare disease with an incidence of 1–3 per 100,000 women, affecting predominantly elderly patients [1,2,3]. Squamous cell carcinoma (SCC) is the most common malignant tumor of the vulva accounting for more than 90% of vulvar malignancies, followed by melanoma and other histological subtypes, including verrucous carcinoma, basal cell carcinoma, adenocarcinomas and Paget’s disease [4].

Vulvar carcinoma is symptomatic in most cases, presenting as a painful vulvar mass or ulcer and causing pruritus, bleeding and discharge. Moreover, a mass in the groin can be reported in the case of inguinal lymph node involvement. Vulvar cancer is confined to the primary site in 59% of cases, regional lymph nodes are involved in 30% and distant metastasis can be found in 6% of cases [5]. Thus, vulvar cancer frequently spreads to the groin, and inguinofemoral lymph node (LN) involvement is the most significant prognostic factor for survival [6,7]. The 5-year survival rate is 86% for localized disease, which decreases to 53% in cases of regional spread [8].

The most widely used staging system in vulvar cancer was created by the International Federation of Gynecology and Obstetrics (FIGO), revised in 2021 [9,10].

Surgery is required for this disease. Locally, radical excision is necessary and, in some cases, associated with plastic reconstruction to repair the tissue defect and reduce postoperative complications. In particular, vulvoperineal surgery can range from very small excisions to extensive surgeries involving neighboring organs. Surgery can be modulated in order to achieve free surgical margins (>8 mm) [11].

It is clear that tumors <2 cm with stromal invasion <1 mm (FIGO stage IA) do not require lymph node surgical staging nor preoperative lymph node imaging, since the risk of metastasis is less than 1%. Indeed, it appears necessary when the disease FIGO stage is IB or II, because the potential risk of metastasis in these cases raises to over 8%, even when inguinal lymph nodes are negative at the clinical examination.

Moreover, groin treatment should be modulated according to clinical characteristics of the tumor and imaging. In particular, a key element of preoperative evaluation to assess groin surgery planning is a precise clinical exam with accurate evaluation of tumor size, number of lesions (focality), distance from midline and infiltration of extravulvar tissues and organs. In fact, multifocality and tumor size >4 cm are currently exclusion criterion from sentinel node procedures, whereas a midline lesion requires a bilateral inguinal approach. In locally advanced disease, evident or suspicious infiltration of deep structures (urethra/bladder, vagina, anus/rectum) requires further investigation with MRI or other specific exams. This would allow to define local spread and guide treatment planning (radical surgery versus chemoradiation up front).

Finally, with regard to the initial assessment of the disease, complete excisional biopsy of the lesions is not recommended, as this will affect groin lymph node treatment. In this case, the data currently available in the literature do not allow a minimally invasive surgical approach.

It is well known that radical groin dissection is associated with a high rate of postoperative morbidity, due to the most common postoperative complications such as wound infection (20–40%), wound breakdown and debilitating lymphedema (30–70%) [12].

In light of the frequent complications of radical lymphadenectomy, which can be very severe and disabling, many researchers tried to understand if less extensive surgery was feasible without impairing prognosis. The first innovation in this field was the introduction of a minimally invasive surgical technique, already widely used in the treatment of breast cancer and skin melanoma: the sentinel node biopsy.

Currently, the sentinel lymph node (SLN) biopsy is recommended for surgical nodal staging in the management of early-stage vulvar cancer, with strict selection criteria: unifocal tumors <4 cm without suspicious groin nodes on clinical examination and preoperative imaging [13]. The use of this technique has resulted in decreased postoperative morbidity without compromising detection of LN metastases [14,15,16]. Some studies were conducted with the aim of extending the indications for the sentinel lymph node biopsy, which could be reached more easily having in hand an adequate screening tool in the preoperative phase [17,18,19].

Therefore, the preoperative assessment of the groin LNs is still essential to customize surgery and avoid unnecessary procedures. In the preoperative assessment, clinical palpation is not particularly sensitive in detecting inguinal lymph node metastases. Data reported in the literature showed that clinical palpation had a sensitivity ranging from 35 to 64% in detecting groin metastasis [20,21,22,23].

Imaging techniques available and approved in the preoperative assessment of baseline staging in vulvar cancer include computed tomography scans (CT), magnetic resonance imaging (MRI) and fluorodeoxyglucose positron emission tomography (FDG PET) [17,22,24,25,26,27,28,29,30,31,32]. In the detection of inguinal lymph node metastases, CT scans have shown a negative predictive value around 75% [33,34,35]. As part of a systematic review, Selman et al. stated that MRI has a pooled sensitivity and specificity of 86 and 87%, respectively, in predicting the groin node status in vulvar cancer. Results consistent with those described by Selman et al. were also reported by Singh et al. (sensitivity of 85.7% and specificity of 82.1%) and Kataoka et al. (sensitivity of 86.7% and specificity of 81.3%). However, it should be noted that the MRI criterion used for groin lymph node metastasis prediction varied between the studies [24,29]. Data reported in the literature on the efficacy of FDG PET to assess LN status in vulvar cancer showed a sensitivity from 50 to 75%, a specificity from 52 to 100% and a negative predictive value of 93% [30,31,32].

The advent of ultrasound with higher transducer frequencies and near focusing with the ability to visualize superficial tissue had resulted in a method for preoperative assessment of superficial inguinal lymph nodes.

In particular, many studies performed on breast cancer have evaluated the role of preoperative ultrasound on superficial lymph nodes, such as in the axilla with sensitivity between 87 and 77% [36,37], whereas specificity ranged from 56 to 93.6% [38].

Its role in evaluating the LN status in vulvar cancer is already approved by the European Society of Gynaecological Oncology’s (ESGO) Guidelines for the Management of Vulvar Cancer [39] but not yet recognized by other guidelines, such as the National Comprehensive Cancer Network (NCCN) guidelines [40], in which pelvic MRI or FDG PET could be considered in the initial workup in vulvar cancer.

The aim of this systematic review and meta-analysis is to determine the efficacy of ultrasound to assess the inguinal lymph nodes in patients with vulvar cancer.

## 2. Materials and Methods

The present work was reported according to the Preferred Reporting Items for Systematic Reviews and Meta-Analyses (PRISMA) statement [41]. The protocol of this systematic review was published on PROSPERO, registration number CRD42021238776.

### 2.1. Research Question

This systematic review aimed at answering the question: What is the efficacy of ultrasound in the evaluation of inguinal lymph nodes in women with all types and all stages of vulvar cancer compared to histological examination? The question was structured according to the PICOS statement as follows:

Population: women with all types (SCC, melanoma, adenocarcinoma, basal cell carcinoma) and all stages of vulvar cancer; 

Intervention: performing ultrasound;

Comparison: histology (biopsy, lymphadenectomy, FNAC, sentinel lymph node);

Outcome: efficacy of ultrasound in the evaluation of inguinal lymph nodes (if available, sensitivity, specificity, positive predictive value, negative predictive value, positive likelihood ratio, negative likelihood ratio and accuracy were reported, or when data were available was calculated).

### 2.2. Search Strategy

To retrieve potential eligible articles, the electronic databases of PubMed, Web of Science and Scopus were searched on 19 October 2020. A search string for PubMed consisting of Medical Subject Headings terms and free text words was developed. Afterwards, this search string was adjusted for use in the other electronic databases. The keywords used includes: “Vulvar Neoplasms”, “Vulvar cancer” “Vulvar carcinoma”, “Vulvar malignancy”, “groin recurrence” OR “groin metastasis” OR “inguinal node metastasis”, “inguinal sentinel lymph node”, “Ultrasonography”, ultrasound “Ultrasonic Diagnosis”, “Echography”, “Sonography”. Search was restricted to articles published in English and regarding humans, without any further restrictions (See Appendix A).

### 2.3. Eligibility Criteria and Selection Process

All study types that reported primary data on the role of ultrasound in the evaluation of groin lymph nodes in vulvar cancer were included in the systematic review. We excluded non-empirical studies, only abstract studies and conference papers. Furthermore, we excluded perspectives, opinions, comments or editorials. All articles retrieved from the search strategy were imported to Rayyan QCRI and duplicates were removed. Two independent reviewers (DV and DZ) selected the identified studies based on the title and abstract. If the topic of the study could not be ascertained from the title/abstract, the full text version was retrieved for evaluation. When we could not find the full texts, corresponding authors were contacted. Disagreement was resolved by consensus.

### 2.4. Data Extraction

Data extraction was conducted by two researchers (DV, DZ), using a dedicated Excel spreadsheet. The following data were extracted and recorded for each included article: first author, year of publication, country, study design, sample size, mean age of the studied population, cancer histologic type and staging, time of ultrasound assessment (preoperatively/during follow up), comparison (histology) number of groins assessed at ultrasound, ultrasound parameters considered as predictive of nodal metastasis andsensitivity (Se), specificity (Sp), positive predictive value (PPV), negative predictive value (NPV), positive likelihood ratio (LR+), negative likelihood ratio (LR-), accuracy (ACC) of ultrasound in detecting groin lymph nodes’ involvement.

### 2.5. Data Synthesis and Analysis

Data retrieved from the included studies were pooled in random-effects meta-analyses. We used STATA to draw statistical graphs and pool statistical measures such as forest plots for Se, Sp, PPV, NPV, LR+, LR− and ACC with corresponding 95% confidence intervals (CI). If these statistics were not given, they were calculated where appropriate data were available.

The hierarchical summary receiver-operating characteristic curve (HSROC) was used for ultrasound accuracy. We used the inconsistency index (I^2^) to estimate the heterogeneity across the included studies. The heterogeneity between studies was considered low if the I^2^ value was <50%. Galbraith’s test and sensitivity analysis were conducted to investigate the impact each study had on the overall estimate and its contribution to Q-statistics [42]. Publication bias was assessed by Begg’s test [43], where *p* < 0.05 indicated significant publication bias and the presentation of funnel plots. Statistical significance was considered at *p* < 0.05. Statistical analyses were performed using the STATA software package v. 15 (Stata Corporation, College Station, TX, USA).

### 2.6. Quality Assessment

The QUADAS-2 (Quality Assessment of Diagnostic Accuracy Studies-2) [44] tool was used to evaluate the quality of the included studies. This tool is composed of four parts: patient selection, index test, reference standard, and flow and timing. Every part was evaluated for bias risk and the first three in terms of concerns regarding applicability. Each part was assigned a score of unclear, high or low.

## 3. Results

### 3.1. Search Strategy

Our search strategy retrieved a total of 926 articles. After the screening and selection process, eight articles [34,45,46,47,48,49,50,51] were deemed pertinent for inclusion in the systematic review and meta-analysis (Figure 1).

### 3.2. Characteristics of the Included Studies

Studies were conducted mostly in the UK (50%) [34,45,47,51], followed by Germany [46], Italy [48], Finland [49] and the Netherlands [50] (each 12.5%) in the time period from 1993 to 2020. Sample size varied from 20 patients [51] to 144 patients [48], with a median age from 67 [50] to 76 [46]. In 62.5% of cases the histological type of cancer was squamous cell carcinoma [34,45,47,49,50], whereas in the rest of the studies all histological types were included. The number of groins examined ranged from 37 [51] to 256 [48]. Moskovic et al. considered oval nodes with substantial hilar fat and minimal lymphoid tissue as benign or reactive, whereas nodes with a more circular or irregular configuration and with loss of central hilar fat as suspicious or malignant. In the study of de Gregorio et al., the absence of fatty hilum in the lymph node, irregular shape, cortical region diameter ≥4 mm and peripheral vascularization were criteria of malignancy (subjective assessment was the index test used for classifying lymph nodes). Hall et al. included lymph node size, shape (long-to-short-axis diameter ratio), preservation of an echogenic hilum, general attenuation and vascularity on Doppler as criteria for nodal evaluation. A rounded shape with a ratio of less than 2, the loss of hilar sinus fat and increasing low attenuation of the cortex were considered suspicious or malignant findings, as was peripheral vascularity on color and power Doppler. In the study of Garganese et al., a specific group of morphological parameters was recorded using a dedicated form to register either the absence or presence of each feature. In particular, the list of recorded parameters included: globular shape; inhomogeneous echostructure; intranodal deposits; hilum anomalies (absence, displacement or interface distortion); cortical thickening; nodal grouping; the presence of a perinodal hyperechoic ring (as a sign of inflammatory perinodal stroma) and cortical interruption (as a sign of extracapsular tumor spread). Additionally, the presence of rich vascularization was assessed. Moreover, the following dimensional parameters were recorded: long (L) and short axis (S) with cortical (C) and medullar (M) thickness of the dominant LN. In addition to the measurement of the parameter size as well, the corresponding ratios were then considered.

Thus, with regard to the L/S ratio, a value <2 was judged as suspicious (correlated to the presence of a rounded morphology and non-elliptical lymph node). Finally, for the C/M ratio, a value >1 was retained as suspicious (as it represented a cortex with thickness equal to or greater than the medullary component of the lymph node).

Makela et al. described the following sonographic criteria for pathologic lymph nodes: nodal hypoechogenicity, absence of internal hilar echoes, greatest diameter over 1.5 cm, roundish shape and thickness ratio length over 1/2. Pouwer et al. considered as suspicious on ultrasound lymph nodes characterized by a short-axis diameter ≥10 mm in oval-shaped lymph nodes or ≥8 mm in circular-shaped lymph nodes with a malignant aspect. Malignant aspects visualized by ultrasound were hilar hypoechogenicity, general attenuation, irregularity of the margin or an abnormal vascular pattern on Doppler. Land et al. assessed the ultrasound lymph node size, shape (long-to-short(L/S)-axis diameter ratio), preservation of echogenic hilum and general attenuation, and vascularity on Doppler. Nodes appearing as oval (L/S ratio greater than 2) with substantial hilar fat and minimal lymphoid tissue were considered benign or reactive, and those with a more circular or irregular configuration and with loss of central hilar fat were considered suspicious or malignant. In all these seven studies, the predictive performance of subjective assessment was considered for the analysis and the histology was considered as gold standard. Finally, in the study of Mohammed et al., all lymph nodes were assessed for size, looking at the maximum length and taking the short axis tangential to this across the maximum diameter of the node. This gave the L/S ratio and the maximum short axis. Lymph nodes with a short axis >8 mm and/or L/S ratio <2 were considered suspicious or malignant. In the study of Mohammed, the predictive performance of the combination of both ultrasound parameters (short axis and L/S ratio) was considered for the analysis and the histology was considered as gold standard.

### 3.3. Quality Assessment

The selection of patients could have introduced bias in 50% of studies, whereas in 37.5% this bias was low and in 12.5% was unclear. The bias from the conduct or interpretation of the index test was assessed to be low in all studies. The bias from the conduct or interpretation of reference standard was unclear in 50% of studies, low in 37.5% and high in only 12.5%. The patients’ flow had a high probability of introducing bias in half of the studies, whereas in the other half this bias was low.

### 3.4. Meta-Analyses

The random-effects model showed a pooled Se and Sp of 0.85 (95% CI: 0.81–0.89) and 0.86 (95% CI: 0.81–0.91), respectively (Figure 2). The analysis for pooled Se, NPV and LR- excluded the study by Power et al. because the effect measure could not be evaluated (Se = 1; NPV = 1; LR− = 0). No heterogeneity among studies was found in the meta-analysis concerning Se. Regarding Sp, there was a high heterogeneity (I^2^ = 85.1% *p*-value < 0.001), which did not decrease (I^2^ = 75.9%; *p*-value < 0.001) even after omitting the study by Garganese et al. that could contribute to the heterogeneity according to the sensitivity analysis. Begg’s test showed a possible publication bias for Sp (*p*-value = 0.01) and Se (*p*-value = 0.03).

There was a pooled PPV and NPV of 0.65 (95% CI: 0.54–0.79) and 0.92 (95% CI: 0.91–0.94), respectively. No heterogeneity was found for NPV (I^2^ = 18.2%; *p*-value = 0.29), whereas there was a high heterogeneity concerning PPV (I^2^ = 92%; *p*-value < 0.001). The sensitivity analysis showed that the study by de Gregorio et al. and Land et al. could contribute to this heterogeneity, but we saw no decrease even after omitting these studies from the analysis (I^2^ = 93.1%; *p*-value < 0.001 and I^2^ = 91.3%; *p*-value < 0.001) (Figure 3). Begg’s test showed no publication bias for PPV (*p*-value = 0.7) and NPV (*p*-value = 0.2).

There was a pooled LR+ and LR− of 6.44 (95% CI: 3.72–11.4) and 0.20 (95% CI: 0.14–0.27), respectively. No heterogeneity was found for LR− (I^2^ = 0.0%; *p*-value = 0.96), whereas there was a high heterogeneity concerning LR+ (I^2^ = 85.7% *p*-value < 0.001). The sensitivity analysis showed that the study by Power et al. could contribute to this heterogeneity, and we saw a decrease in heterogeneity omitting this study from the analysis (I^2^ = 58.9%; *p*-value = 0.03) (Figure 4). Begg’s test showed no publication bias in both cases (*p*-value = 0.3 and *p*-value = 0.7, respectively).

The pooled accuracy was 0.85 (95% CI: 0.80–0.91) with a high heterogeneity among studies (I^2^ = 85.8%; *p*-value < 0.001), which slightly decreased (I^2^ = 64.8%; *p*-value < 0.009) after omitting the study by Power et al. that could contribute to the heterogeneity according to the sensitivity analysis (Figure 5). Begg’s test showed no publication bias (*p*-value = 0.3) (See Appendix A for the funnel plot).

The hierarchical summary receiver-operating characteristic curve (HSROC) with summary point, summary estimates, 95% confidence region and 95% prediction region for all included studies of ultrasound evaluating inguinal lymph nodes in patients with vulvar cancer is shown in Figure 6.

## 4. Discussion

This review includes data from 437 women and 914 groins from eight studies (Moskovic 1999, de Gregorio 2013, Hall 2003, Garganese 2020, Makela 1993, Pouwer 2018, Land 2006, Mohammed 2000). To the best of our knowledge, the present study represents the best available summary of existing research to date. All the studies included, except one, were retrospective and ultrasound criteria for groin lymph node metastasis prediction varied among them (see Results, *Characteristics of the Included Studies)*. The quality assessment of the included studies showed that most of their results could be impaired by selection bias and the patients’ flow. In general, the index test and the reference test used had low probability of introducing bias. Despite these limitations, this review has demonstrated that ultrasound examination has high sensitivity (Se 0.85; 95% CI: 0.81–0.89) and high negative predictive value (NPV 0.92; 95% CI: 0.91–0.94) in predicting nodal status in vulvar cancer. No heterogeneity among studies was found in the meta-analysis concerning Se and NPV. In the literature, some studies used MRI to evaluate lymph node metastases in vulvar cancer. In particular, Serrado et al. in 2019 reported widely varying sensitivities and specificities, ranging from 40% to 89% and from 81% to 100%, respectively [52]. Shetty et al., as well as Kataoka et al., described a short-to-long-axis ratio greater than 0.75 as the parameter with the greatest overall accuracy (85% and 84.8%) and the presence of necrosis as the most specific feature [25,53].

Moreover, Sohaib et al., on a groin-based analysis from 22 patients, found a sensitivity and specificity of 40% and 97%, respectively, using 10 mm short-axis diameter cutoff [54]. Bipat et al., using the short-axis diameter >8 mm as a predictive parameter, showed a sensitivity of 52%, specificity of 88% and NPV of 89% [55]. Singh et al., providing specific diagnostic criteria (short-axis diameter >10 mm; irregular or rounded shape; increased signal intensity on T2 sequences), on a per-groin analysis, showed sensitivity of 85.7%, specificity of 82.1% and NPV of 93.9% [22].

In summary, several different MRI criteria have been proposed as the most accurate to identify metastatic groin lymph nodes. The most common criterion is the short-axis diameter, although its low sensitivity of 40–50%, depending on the lymph node location and the cut-off value, must be considered. Other suggested criteria were the irregularity of the lymph node contour, the ratio of the long-to-short-axis diameter and the presence of necrosis in the lymph node.

Moreover, for CT scans the main criterion of suspicion is mainly represented by a short axis >1 cm; therefore, metastatic nodes with a short axis less than 1 cm could be easily missed.

In fact, a CT scan is not routinely recommended for the assessment of inguinal lymph node status in vulvar cancer due to its low sensitivity of 58–60% in detecting groin metastasis with a PPV ranging from 38 to 58% and a NPV of 75–96% [52]. In addition, the role of 18F-FDG PET/CT in preoperative assessment of inguinal status in vulvar cancer were investigated and a sensitivity ranging from 50% to 100%, a specificity ranging from 67% to 100% and a negative predictive value higher than 90% were reported [52].

Triumbari et al. included in a recent systematic review and meta-analysis seven articles which evaluated the diagnostic performance of preoperative 18F-FDG PET/CT for lymph node staging [32]. The authors revealed that a negative preoperative PET/CT scan may exclude groin metastases in at least early-stage vulvar cancer patients, otherwise a positive PET/CT result should be interpreted with caution. Rufini et al. in their recent retrospective large monocentric study analyzed 160 patients and 338 LN sites, confirming that 18F-FDG PET/CT showed good values of sensitivity (85,6%) and NPV in discriminating metastatic from non-metastatic LNs [30]. The present review supports the use of ultrasound to assess inguinal lymph nodes in patients with vulvar cancer considering the high predictive performance and the advantages of this method (i.e., cost effective, easy to learn, easy to perform, not requiring use of radiation or contrast agents). This finding is clinically relevant to optimize the management of patients with vulvar cancer, especially in institutions where more expensive radiological investigations (e.g., CT scans/MRI or PET/CT scans) are not easily accessible due to their costs. In particular, ultrasound could represent the first imaging step to select patients negative for macrometastases and potentially eligible for less invasive surgical treatments. Prospective multicenter studies are necessary to confirm the ability of ultrasound in detecting metastatic lymph nodes and to define the best clinical strategy in this setting.

Actually, on the basis of the most relevant studies published, both the ACR [30] ACR and ESUR [31] ESUR include ultrasound as an adequate imaging technique for lymph node staging, although mainly considered as a guide for cytological/histological sampling [56,57].

Therefore, it could be of interest to investigate how invasive diagnostics as fine-needle aspiration or core biopsy can improve ultrasound performance. In fact, in the study of Mohammed [51], fine-needle aspiration cytology (FNAC) was evaluated as the completion of a suspicion ultrasound due to a L/S ratio <2 or short axis >8 mm. In this case, FNAC has not been shown to be effective in improving ultrasound performance due to poor diagnostic power (high non-diagnostic and false negative rate). In 2019 Angelico et al. retrospectively evaluated 43 FNAC specimens from suspicious lymph nodes at ultrasound; cytology agreed with histology in 86.04% of cases [58]. No data are available regarding core biopsy in groin nodes for patients with vulvar cancer. Several studies have been conducted regarding the role of core biopsy and ultrasound in the preoperative staging of axillary nodes in breast cancer. What clearly emerges is the high level of PPV ranging from 79 to 100% [59,60]. In 2018 Balasubramanian et al. [61] demonstrated how US core biopsy was superior to US-FNA in diagnosing axillary nodal metastases.

It is evident that the rarity of the disease associated with the lack of a standardized ultrasound nomenclature to describe morphological features of the lymph nodes has made it difficult to compare the results of different studies and find reliable ultrasound criteria to identify suspicious lymph nodes.

The recent publication of the consensus of the VITA (Vulvar International Tumor Analysis) group on terms, definitions and measurements described sonographic features of lymph nodes in vulvar cancer [62], overcoming the lack of standardized ultrasound nomenclature.

To have a shared terminology, in fact, represents the first step towards the possibility of obtaining studies with a greater opportunity for comparison. In addition, this will allow us to conduct more easily collaborative studies and open up new perspectives on the development of mathematical models, radiomics and translational analyses and translational analyses.

## 5. Conclusions

Ultrasound should be considered as an accurate tool for assessment of nodal status in vulvar cancer in order to allow patients with low risk of nodal involvement to receive less invasive surgeries (SNB), thus resulting in less postoperative complications of unnecessary lymphadenectomy. Further prospective multicenter studies with homogeneous criteria to define suspicious or certain malignant LNs are desirable to confirm these findings. In this context, it is important for future studies to pay special attention to patient selection in order to avoid any selection bias.

## Figures and Tables

**Figure 1 cancers-14-03082-f001:**
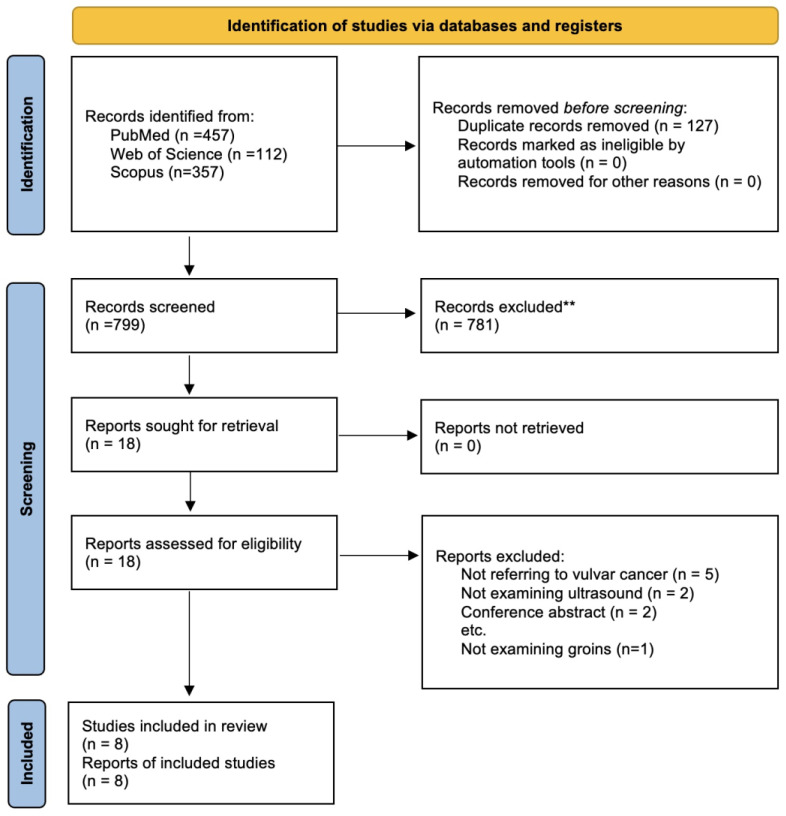
Flow diagram of the screening and selection process (from Page et al., 2021).

**Figure 2 cancers-14-03082-f002:**
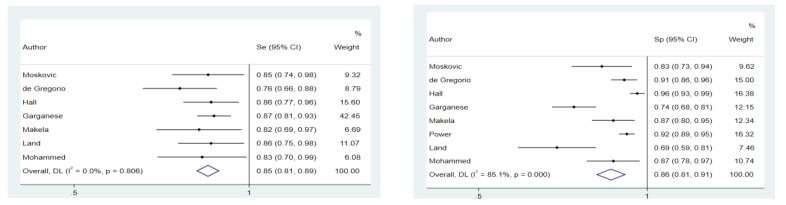
Forest plot for sensitivity and specificity.

**Figure 3 cancers-14-03082-f003:**
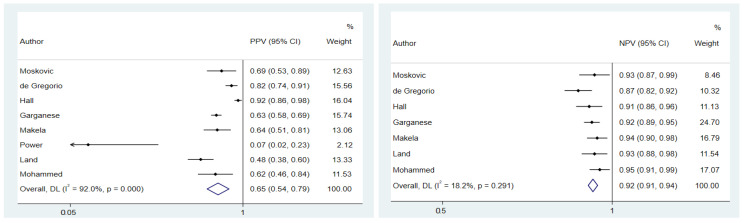
Forest plot for positive predictive value and negative predictive value.

**Figure 4 cancers-14-03082-f004:**
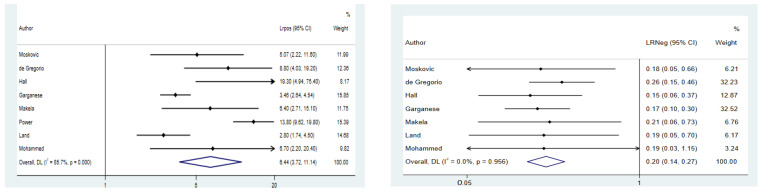
Forest plot for positive likelihood ratio and negative likelihood ratio.

**Figure 5 cancers-14-03082-f005:**
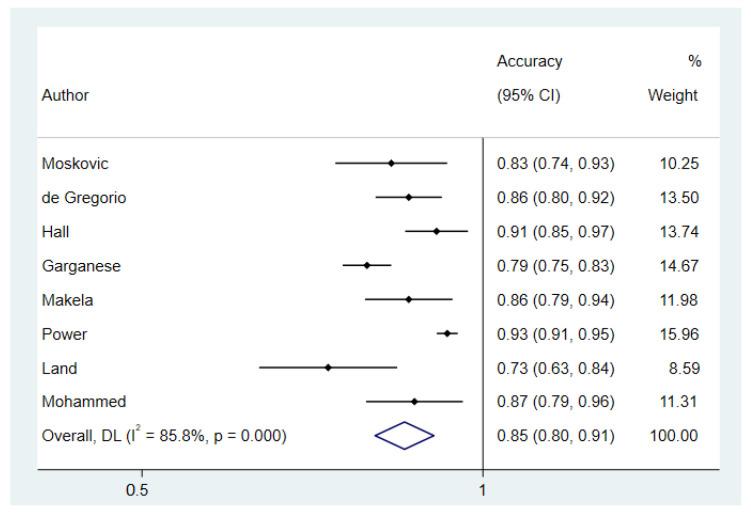
Accuracy.

**Figure 6 cancers-14-03082-f006:**
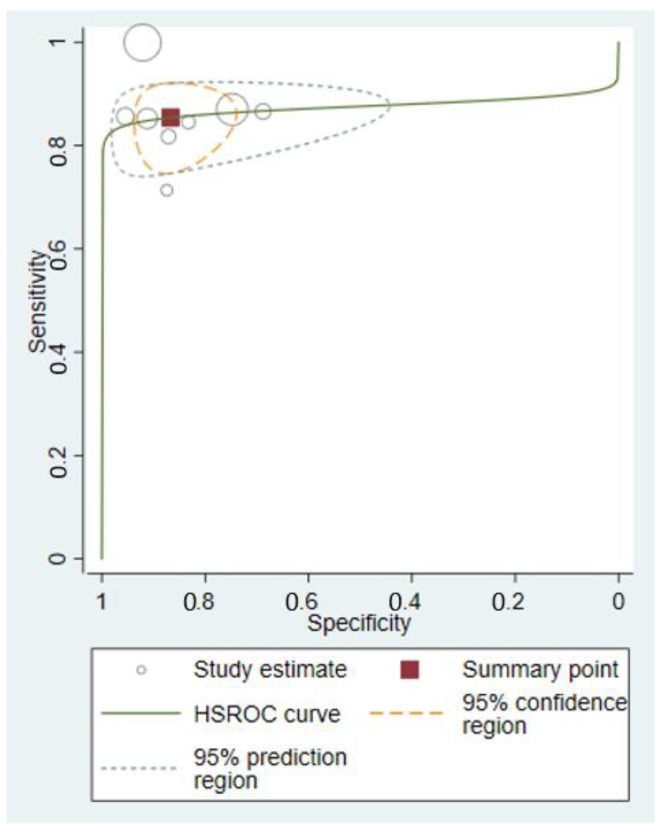
Hierarchical summary receiver-operating characteristic curve (HSROC).

## Data Availability

The data presented in this study are available on reasonable request from the corresponding author.

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
