# Peer review of "The Role of Ultrasound in the Evaluation of Inguinal Lymph Nodes in Patients with Vulvar Cancer: A Systematic Review and Meta-Analysis"

_cancers, 2022, doi:10.3390/cancers14133082_

Round 1

Reviewer 1 Report

It is a very interesting paper dealing with the future of vulvar cancer diagnostica approach.

I suggest few comments

Line 6 Is "Diagnostic power" a correct term ? (the readers are more familiar with sensitivity, specificity ....)

Line 7-9 heavy to follo: please rephrase

Line 14 "efficacy" or sensitivity, specificity ?

Line 35-36 please a synonymous of lump

Line 39 Cite reference

Line 49 Reference (ex.  doi: 10.1111/jog.14962. Epub 2021 Aug 8. PMID: 34365709. or  doi: 10.3802/jgo.2018.29.e61. Epub 2018 Apr 13. PMID: 30022627; PMCID: PMC6078886.)

Author Response

Line 6 Is "Diagnostic power" a correct term? (The readers are more familiar with sensitivity, specificity ….)

REPLY: It has been changed in the text with “accuracy”.

Line 7-9 heavy to follo: please rephrase

REPLY: It has been changed in the text as required.

” To date, MRI showed a great diagnostic accuracy on defining disease extension to soft tissue and deep organs. . At present, regarding inguinal nodes study, PET/CT scan has shown a high negative predictive value although in the presence of a suspicious/positive report it should be taken with caution. We report the results of a study aimed to investigate the role of groin ultrasound in the assessment of lymph nodal status in vulvar cancer.”

Line 14 "efficacy" or sensitivity, specificity?

REPLY: In our purpose the term “efficacy” is correct. In fact, with this term, we intend the primary clinical study objective. Sensitivity and specificity, cited later in the text are intended as outcome measure to describe the study results.

Line 35-36 please a synonymous of lump

REPLY: It has been changed in the text with “mass”.

Line 39 Cite reference

REPLY: We agree with the Reviewer. It has been added in the text, as suggested.

Line 49 Reference (ex.  doi: 10.1111/jog.14962. Epub 2021 Aug 8. PMID: 34365709. or  doi: 10.3802/jgo.2018.29.e61. Epub 2018 Apr 13. PMID: 30022627; PMCID: PMC6078886.)

REPLY: We agree with the Reviewer. It has been added in the text, as suggested.

Reviewer 2 Report

Overall the article is well written and well researched with detailed and appropriate referencing.

There are a few suggestions for your consideration:

1.  In the Introduction, line 52, "It is well known that radical groin dis
section is associated to high rate of post-operative morbidity, due to the most common postoperative complications such as wound infection (20-40%), wound breakdown, and debilitating lymphedema (30-70%) (10). "

Instead could the sentence read -It is well known that radical groin dis
section is associated with a high rate of post-operative morbidity, due to the most common postoperative complications such as wound infection (20-40%), wound breakdown, and debilitating lymphedema (30-70%) (10).

2.  Regarding the 2.1 Research question
"This systematic review aimed at answering the question: Which is the efficacy of ultrasound in the evaluation of inguinal lymph nodes in women with all types and all stages of vulvar cancer compared to the histological examination? "

Instead could the sentence read - What is the efficacy of ultrasound in the evaluation of inguinal lymph nodes in women with all types and all stages of vulvar cancer compared to the histological examination? "

3. In the Discussion and in reference to line 313:

"Moreover, at CT scan the main criterion of suspicion is mainly represented by short axis>1cm, therefore metastatic nodes whit a short axis minor than 1cm could be easily missed. "

Could the sentence please read - Moreover, at CT scan the main criterion of suspicion is mainly represented by short axis>1cm, therefore metastatic nodes with a short axis minor than 1cm could be easily missed.

4. In the Discussion and in reference to lines 340 and 341:

"Moreover, it could be of interest to investigate how invasive diagnostics as fine needle aspiration or core biopsy can improve ultrasound performance. In fact, in the study of Mohammed (54) fine needle aspiration citology (FNAC) was evaluated as a completion of a suspicion ultrasound due to L/S ratio<2 or short axis>8mm."

Could the sentence instead read - Moreover, it could be of interest to investigate how invasive diagnostics as fine needle aspiration or core biopsy can improve ultrasound performance. In fact, in the study of Mohammed (54) fine needle aspiration cytology (FNAC) was evaluated as a completion of a suspicious ultrasound due to L/S ratio<2 or short axis>8mm."

5. In the Discussion and in reference to line 346:

"Several are the studies conducted on breast cancer regarding the role of core biopsy and ultrasound in preoperative staging of axillary nodes."

Could this sentence please be changes as grammatically the sentence is difficult to read clearly.

In the Discussion or Conclusion there is a need to mention the importance of inguinal lymph node ultrasound as part of the metastatic work up for biopsy proven vulvar cancer, especially in institutions when more expensive radiological investigations eg. CT scans/MRI or PET scans are not made readily available due to their cost.

Author Response

  1. In the Introduction, line 52, "It is well known that radical groin dis

section is associated to high rate of post-operative morbidity, due to the most common postoperative complications such as wound infection (20-40%), wound breakdown, and debilitating lymphedema (30-70%) (10). "

Instead, could the sentence read -It is well known that radical groin dissection is associated with a high rate of post-operative morbidity, due to the most common postoperative complications such as wound infection (20-40%), wound breakdown, and debilitating lymphedema (30-70%) (10).

REPLY: It has been changed in the text, as suggested.

  1. Regarding the 2.1 Research question

"This systematic review aimed at answering the question: Which is the efficacy of ultrasound in the evaluation of inguinal lymph nodes in women with all types and all stages of vulvar cancer compared to the histological examination? "

Instead could the sentence read - What is the efficacy of ultrasound in the evaluation of inguinal lymph nodes in women with all types and all stages of vulvar cancer compared to the histological examination? “

REPLY: We agree with the Reviewer. It has been changed in the text, as suggested.

  1. In the Discussion and in reference to line 313:

"Moreover, at CT scan the main criterion of suspicion is mainly represented by short axis>1cm, therefore metastatic nodes whit a short axis minor than 1cm could be easily missed. "

Could the sentence please read - Moreover, at CT scan the main criterion of suspicion is mainly represented by short axis>1cm, therefore metastatic nodes with a short axis minor than 1cm could be easily missed.

REPLY: We agree with the Reviewer. It has been changed in the text.

  1. In the Discussion and in reference to lines 340 and 341:

"Moreover, it could be of interest to investigate how invasive diagnostics as fine needle aspiration or core biopsy can improve ultrasound performance. In fact, in the study of Mohammed (54) fine needle aspiration citology (FNAC) was evaluated as a completion of a suspicion ultrasound due to L/S ratio<2 or short axis>8mm."

Could the sentence instead read - Moreover, it could be of interest to investigate how invasive diagnostics as fine needle aspiration or core biopsy can improve ultrasound performance. In fact, in the study of Mohammed (54) fine needle aspiration cytology (FNAC) was evaluated as a completion of a suspicious ultrasound due to L/S ratio<2 or short axis>8mm.”

REPLY: We agree with the Reviewer. It has been changed as suggested.

  1. In the Discussion and in reference to line 346:

"Several are the studies conducted on breast cancer regarding the role of core biopsy and ultrasound in preoperative staging of axillary nodes."

Could this sentence please be changed as grammatically the sentence is difficult to read clearly.

REPLY: We agree with the Reviewer. It has been changed in the text, as suggested.

“Several studies have been conducted regarding the role of core biopsy and ultrasound in the preoperative staging of axillary nodes in breast cancer.”

In the Discussion or Conclusion there is a need to mention the importance of inguinal lymph node ultrasound as part of the metastatic work up for biopsy proven vulvar cancer, especially in institutions when more expensive radiological investigations eg. CT scans/MRI or PET scans are not made readily available due to their cost.

REPLY: It has been added in the text, as suggested.

“This finding is clinically relevant to optimize the management of patients with vulvar cancer, especially in Institutions where more expensive radiological investigations (eg. CT scans/MRI or PET/CT scans) are not easily accessible due to their costs.”
